# Copula-like Variational Inference

**Marcel Hirt**
Department of Statistical Science
University College of London, UK
`marcel.hirt.16@ucl.ac.uk`

**Petros Dellaportas**
Department of Statistical Science
University College of London, UK
Department of Statistics
Athens University of Economics and Business, Greece
and The Alan Turing Institute, UK

**Alain Durmus**
CMLA
École normale supérieure Paris-Saclay,
CNRS, Université Paris-Saclay, 94235 Cachan, France.
`alain.durmus@cmla.ens-cachan.fr`

## Abstract

This paper considers a new family of variational distributions motivated by Sklar's theorem. This family is based on new copula-like densities on the hypercube with non-uniform marginals which can be sampled efficiently, *i.e.* with a complexity linear in the dimension $d$ of the state space. Then, the proposed variational densities that we suggest can be seen as arising from these copula-like densities used as base distributions on the hypercube with Gaussian quantile functions and sparse rotation matrices as normalizing flows. The latter correspond to a rotation of the marginals with complexity $\mathcal{O}(d \log d)$. We provide some empirical evidence that such a variational family can also approximate non-Gaussian posteriors and can be beneficial compared to Gaussian approximations. Our method performs largely comparably to state-of-the-art variational approximations on standard regression and classification benchmarks for Bayesian Neural Networks.

## 1 Introduction

Variational inference [29, 68, 4] aims at performing Bayesian inference by approximating an intractable posterior density $\pi$ with respect to the Lebesgue measure on $\mathbb{R}^d$, based on a family of distributions which can be easily sampled from. More precisely, this kind of inference posits some variational family Q of densities $(q_\xi)_{\xi \in \Xi}$ with respect to the Lebesgue measure and intends to find a good approximation $q_{\xi^\star}$ belonging to Q by minimizing the Kullback-Leibler (KL) with respect to $\pi$ over Q, *i.e.* $\xi^\star \approx \arg\min_{\xi \in \Xi} \mathrm{KL}(q_\xi | \pi)$. Further, suppose that $\pi(x) = \mathrm{e}^{-U(x)}/\mathrm{Z}$ with $U \colon \mathbb{R}^d \to \mathbb{R}$ measurable and $\mathrm{Z} = \int_{\mathbb{R}^d} \mathrm{e}^{-U(x)} \mathrm{d}x < \infty$ is an unknown normalising constant. Then, for any $\xi \in \Xi$,

$$\mathrm{KL}(q_\xi | \pi) = - \int_{\mathbb{R}^d} q_\xi(x) \log \frac{\pi(x)}{q_\xi(x)} \mathrm{d}x = -\mathbb{E}_{q_\xi(x)} \left[ -U(x) - \log q_\xi(x) \right] + \log \mathrm{Z} . \qquad (1)$$

Since Z does not depend on $q_\xi$, minimizing $\xi \mapsto \mathrm{KL}(q_\xi | \pi)$ is equivalent to maximizing $\xi \mapsto \log \mathrm{Z} - \mathrm{KL}(q_\xi | \pi)$. A standard example is Bayesian inference over latent variables $x$ having a prior density $\pi_0$ for a given likelihood function $L(y^{1:n}|x)$ and $n$ observations $y^{1:n} = (y^1, \dots, y^n)$. The target density is the posterior $p(x|y^{1:n})$ with $U(x) = -\log \pi_0(x) - \log L(y^{1:n}|x)$ and the objective that is commonly maximized,

$$\mathcal{L}(\xi) = \mathbb{E}_{q_\xi(x)} \left[ \log \pi_0(x) + \log L(y^{1:n}|x) - \log q_\xi(x) \right] \qquad (2)$$

is called a variational lower bound or ELBO. One of the main features of variational inference methods is their ability to be scaled to large datasets using stochastic approximation methods [24] and applied to non-conjugate models by using Monte Carlo estimators of the gradient [57, 35, 60, 63, 38]. However, the approximation quality hinges on the expressiveness of the distributions in Q and restrictive assumptions on the variational family that allow for efficient computations such as mean-field families, tend to be too restrictive to recover the target distribution. Constructing an approximation family Q that is both flexible to closely approximate the density of interest and at the same time computationally efficient has been an ongoing challenge. Much effort has been dedicated to find flexible and rich enough variational approximations, for instance by assuming a Gaussian approximation with different types of covariance matrices. For example, full-rank covariance matrices have been considered in [1, 28, 63] and low-rank perturbations of diagonal matrices in [1, 46, 53, 47]. Furthermore, covariance matrices with a Kronecker structure have been proposed in [42, 70]. Besides, more complex variational families have been suggested: such as mixture models [18, 22, 46, 40, 39], implicit models [45, 26, 67, 69, 64], where the density of the variational distribution is intractable. Finally, variational inference based on normalizing flows has been developed in [59, 34, 65, 43, 3]. As a special case and motivated by Sklar's theorem [62], variational inference based on families of copula densities and one-dimensional marginal distributions have been considered by [66] where it is assumed that the copula is a vine copula [2] and by [23] where the copula is assumed to be a Gaussian copula together with non-parametric marginals using Bernstein polynomials. Recall that $c : [0,1]^d \to \mathbb{R}_+$ is a copula if and only if its marginals are uniform on $[0,1]$, *i.e.* $\int_{[0,1]^{d-1}} c(u_1, \ldots, u_d) \mathrm{d}u_1 \cdots \mathrm{d}u_{i-1} \mathrm{d}u_{i+1} \cdots \mathrm{d}u_d = \mathbb{1}_{[0,1]}(u_i)$ for any $i \in \{1, \ldots, d\}$ and $u_i \in \mathbb{R}$. In the present work, we pursue these ideas but propose instead of using a family of copula densities, simply a family of densities $\{c_\theta : [0,1]^d \to \mathbb{R}_+\}_{\theta \in \Theta}$ on the hypercube $[0,1]^d$. This idea is motivated from the fact that we are able to provide such a family which is both flexible and allow efficient computations.

The paper is organised as follow. In Section 2, we recall how one can sample more expressive distributions and compute their densities using a sequence of bijective and continuously differentiable transformations. In particular, we illustrate how to apply this idea in order to sample from a target density by first sampling a random variable $U$ from its copula density $c$ and then applying the marginal quantile function to each component of $U$. A new family of copula-like densities on the hypercube is constructed in Section 3 that allow for some flexibility in their dependence structure, while enjoying linear complexity in the dimension of the state space for generating samples and evaluating log-densities. A flexible variational distribution on $\mathbb{R}^d$ is introduced in Section 4 by sampling from such a copula-like density and then applying a sequence of transformations that include $\frac{1}{2} d \log d$ rotations over pairs of coordinates. We illustrate in Section 6 that for some target densities arising for instance as the posterior in a logistic regression model, the proposed density allows for a better approximation as measured by the KL-divergence compared to a Gaussian density. We conclude with applying the proposed methodology on Bayesian Neural Network models.

## 2   Variational Inference and Copulas

In order to obtain expressive variational distributions, the variational densities can be transformed through a sequence of invertible mappings, termed normalizing flows [60]. To be more specific, assume a series $\{\mathscr{T}_t : \mathbb{R}^d \to \mathbb{R}^d\}_{t=1}^T$ of $\mathrm{C}^1$-diffeomorphisms and a sample $X_0 \sim q_0$, where $q_0$ is a density function on $\mathbb{R}^d$. Then the random variable $X_T = \mathscr{T}_T \circ \mathscr{T}_{T-1} \circ \cdots \circ \mathscr{T}_1(X_0)$ has a density $q_T$ that satisfies

$$\log q_T(x_T) = \log q_0(x) - \sum_{t=1}^T \log \det \left| \frac{\partial \mathscr{T}_t(x_t)}{\partial x_t} \right|, \tag{3}$$

with $x_t = \mathscr{T}_t \circ \mathscr{T}_{t-1} \circ \cdots \circ \mathscr{T}_1(x)$. To allow for scalable inferences with such densities, the transformations $\mathscr{T}_t$ must be chosen so that the determinant of their Jacobians can be computed efficiently. One possibility that satisfies this requirement is to choose volume-preserving flows that have a Jacobian-determinant of one. This can be achieved by considering transformations $\mathscr{T}_t \colon x \mapsto H_t x$ where $H_t$ is an orthogonal matrix as proposed in [65] using a Householder-projection matrix $H_t$.

An alternative construction of the same form can be used to construct a density using Sklar's theorem [62, 48]. It establishes that given a target density $\pi$ on $(\mathbb{R}^d, \mathcal{B}(\mathbb{R}^d))$, there exists a continuous function $C\colon [0,1]^d \to [0,1]$ and a probability space supporting a random variable $U = (U_1, \ldots, U_d)$ valued in $[0,1]^d$, such that for any $x \in \mathbb{R}^d$, and $u \in [0,1]^d$,

$$\mathbb{P}\left(U_1 \leqslant u_1, \cdots, U_d \leqslant u_d\right) = C(u_1, \cdots, u_d), \quad \int_{-\infty}^{x_1} \ldots \int_{-\infty}^{x_d} \pi(t)\mathrm{d}t = C(F_1(x_1), \ldots, F_d(x_d)) \tag{4}$$

where for any $i \in \{1, \ldots, d\}$, $F_i$ is the cumulative distribution function associated with $\pi_i$, so for any $x_i \in \mathbb{R}$, $F_i(x_i) = \int_{-\infty}^{x_i} \pi_i(t_i)\mathrm{d}t_i$ and $\pi_i$ is the $i^{\text{th}}$ marginal of $\pi$, so for any $x_i \in \mathbb{R}$, $\pi_i(x_i) = \int_{\mathbb{R}^{d-1}} \pi(x)\mathrm{d}x_1 \cdots \mathrm{d}x_{i-1}\mathrm{d}x_{i+1} \cdots \mathrm{d}x_d$. To illustrate how one can obtain such a continuous function $C$ and random variable $U$, recall that $\pi_i$ is assumed to be absolutely continuous with respect to the Lebesgue measure. Then for $(X_1, \ldots, X_d) \sim \pi$, the random variable $U = \mathscr{G}^{-1}(X) = (F_1(X_1), \ldots, F_d(X_d))$, where $\mathscr{G}\colon [0,1]^d \to \mathbb{R}^d$, with

$$\mathscr{G}\colon u \mapsto (F_1^{-1}(u_1), \ldots, F_d^{-1}(u_d)), \tag{5}$$

follows a law on the hypercube with uniform marginals. It can be readily shown that the cumulative distribution function $C$ of $U$ is continuous and satisfies (4). Note that taking the derivative of (4) yields

$$\pi(x) = c(F_1(x_1), \ldots, F_d(x_d)) \prod_{i=1}^{d} \pi_i(x_i) \,,$$

where $c(u_1, \ldots, u_d) = \frac{\partial}{\partial u_1} \cdots \frac{\partial}{\partial u_d} C(u_1, \ldots, u_d)$ is a copula density function by definition of $C$. One possibility to approximate a target density $\pi$ is then to consider a parametric family of copula density functions $(c_\theta)_{\theta \in \Theta}$ for $\Theta \in \mathbb{R}^{p_c}$ and one parametric family of a $d$-dimensional vector of density functions $(f_1, \ldots, f_d)_{\phi \in \Phi}\colon \mathbb{R}^d \to \mathbb{R}^d$ for $\Phi \subset \mathbb{R}^{p_f}$, and try to estimate $\theta \in \Theta$ and $\phi \in \Phi$ to get a good approximation of $\pi$ via variational Bayesian methods. This idea was proposed by [23] and [66], where Gaussian and vine copulas were used, respectively. The main hurdle for using such family is their computational cost which can be prohibitive since the dimension of $\Theta$ is of order $d^2$. We remark that for latent Gaussian models with certain likelihood functions, a Gaussian variational approximation can scale linearly in the number of observations by using dual variables, see [54, 31].

## 3 Copula-like Density

In this paper, we consider another approach which relies on a copula-like density function on $[0,1]^d$. Indeed, instead of an exact copula density function on $[0,1]^d$ with uniform marginals, we consider simply a density function on $[0,1]^d$ which allows to have a certain degree of freedom in the number of parameters we want to use. The family of copula-like densities that we consider is given by

$$c_\theta(v_1, \ldots, v_d) = \frac{\mathbf{\Gamma}(\alpha^*)}{\mathrm{B}(a,b)} \left[ \prod_{\ell=1}^{d} \left\{ \frac{v_\ell^{\alpha_\ell - 1}}{\mathbf{\Gamma}(\alpha_\ell)} \right\} \right] (v^*)^{-\alpha^*} \cdot \left( \max_{i \in \{1,\ldots,d\}} v_i \right)^a \left[ \left( 1 - \max_{i \in \{1,\ldots,d\}} v_i \right)^{b-1} \right], \tag{6}$$

with the notation $v^* = \sum_{i=1}^{d} v_i$ and $\alpha^* = \sum_{i=1}^{d} \alpha_i$. Therefore $\theta = (a, b, (\alpha_i)_{i \in \{1,\ldots,d\}}) \in (\mathbb{R}_+^* \times \mathbb{R}_+^* \times (\mathbb{R}_+^*)^d) = \Theta$. The following probabilistic construction is proven in Appendix A to allow for efficient sampling from the proposed copula-like density.

**Proposition 1.** *Let $\theta \in \Theta$ and suppose that*

1. $(W_1, \ldots, W_d) \sim Dirichlet(\alpha_1, \ldots, \alpha_d)$;

2. $G \sim Beta(a,b)$;

3. $(V_1, \ldots, V_d) = (GW_1/W^*, \ldots, GW_d/W^*)$, *where* $W^* = \max_{i \in \{1,\ldots,d\}} W_i$.

*Then the distribution of $(V_1, \ldots, V_d)$ has density with respect to the Lebesgue measure given by (6).*

The proposed distribution builds up on Beta distributions, as they are the marginals of the Dirichlet distributed random variable $W \sim \mathrm{Dir}(\alpha)$, which is then multiplied with an independent random variable $G \sim \mathrm{Beta}(a, b)$. The resulting random variable $Y = WG$ follows a Beta-Liouville distribution, which allows to account for negative dependence, inherited from the Dirichlet distribution through a Beta stick-breaking construction, as well as positive dependence via a common Beta-factor. More precisely, one obtains

$$\mathrm{Cor}(Y_i, Y_j) = c_{ij} \left( \frac{\mathbb{E}[G^2]}{\alpha^\star + 1} - \frac{\mathbb{E}[G]^2}{\alpha^\star} \right) ,$$

for some $c_{ij} > 0$ and $\alpha^\star = \sum_{k=1}^{d} \alpha_k$, cf. [13]. Proposition 1 shows that one can transform the Beta-Liouville distribution living within the simplex to one that has support on the full hypercube, while also allowing for efficient sampling and log-density evaluations.

Now note that also $V^- = (1 - V_1, \ldots 1 - V_d)$ is a sample on the hypercube if $V \sim c_\theta$, as is the convex combination $U = (U_1, \ldots, U_d)$, where $U_i = \delta_i V_i + (1 - \delta_i)(1 - V_i)$ for any $\delta \in [0, 1]^d$. Put differently, we can write $U = \mathscr{H}(V)$, where

$$\mathscr{H} : v \mapsto (1 - \delta) \, \mathrm{Id} + \{\mathrm{diag}(2\delta) - \mathrm{Id}\} v , \tag{7}$$

and $\mathrm{Id}$ is the identity operator. It is straightforward to see that $\mathscr{H}$ is a $\mathrm{C}^1$-diffeomorphism for $\delta \in ([0, 1] \backslash \{0.5\})^d$ from the hypercube into $I_1 \times \cdots \times I_d$, where $I_i = [\delta_i, 1 - \delta_i]$ if $\delta_i \in [0, 0.5)$ and $I_i = [1 - \delta_i, \delta_i]$ if $\delta_i \in (0.5, 1]$. Note that the Jacobian-determinant of $\mathscr{H}$ is efficiently computable and is simply equal to $|\prod_{i=1}^{d}(2\delta_i - 1)|$ for $\delta \in [0, 1]^d$.

We suggest to take initially at random $\delta \in [0, 1]^d$ for the transformation $\mathscr{H}$ such that

$$\mathbb{P}(\delta_i = \epsilon) = p \quad \text{and} \quad \mathbb{P}(\delta_i = 1 - \epsilon) = 1 - p \tag{8}$$

with $p, \epsilon \in (0, 1)$. In our experiments, we set $\epsilon = 0.01$ and $p = 1/2$. We found that choosing a different (large enough) value of $\epsilon$ tends to yield no large difference, as this choice will get balanced by a different value of the standard deviation of the Gaussian marginal transformation. The motivation to consider $U = \mathscr{H}(V)$ with $V \sim c_\theta$ was first numerical stability since we need to compute quantile functions only on the interval $[\epsilon, 1 - \epsilon]$ using this transformation. Second, this transformation can increase the flexibility of our proposed family. We found empirically that the components of $V \sim c_\theta$ tend to be non-negative in higher dimensions. However, using sometimes (more) the antithetic component of $V$ by considering $U = \mathscr{H}(V)$, the transformed density can also describe negative dependencies in high dimensions. What comes to mind to obtain a flexible density is then to either optimize over the parameter $\delta$ parametrising the transformation $\mathscr{H}$ or considering $\delta$ as an auxiliary variable in the variational density, resorting to techniques developed for such hierarchical families, see for instance [58, 69, 64]. However, this proved challenging in an initial attempt, since for $\delta_i = 0.5$, the transformation $\mathscr{H}$ becomes non-invertible, while restricting $\delta$ on say $\delta \in \{\epsilon, 1 - \epsilon\}^d$, $\epsilon \approx 0$, seemed less easy to optimize. Consequently, we keep $\delta$ fixed after sampling it initially according to (8). A sensible choice was $p = 1/2$ since it leads to a balanced proportion of components of $\delta$ equal to $\epsilon$ and $1 - \epsilon$. However, the sampled value of $\delta$ might not be optimal and we illustrate in the next section how the variational density can be made more flexible.

## 4 Rotated Variational Density

We propose to apply rotations to the marginals in order to improve on the initial orientation that results from the sampled values of $\delta$. Rotated copulas have been used before in low dimensions, see for instance [36], however, the set of orthogonal matrices has $d(d-1)/2$ free parameters. We reduce the number of free parameters by considering only rotation matrices $\mathcal{R}_d$ that are given as a product of $d/2 \log d$ Givens rotations, following the FFT-style butterfly-architecture proposed in [16], see also [44] and [49] where such an architecture was used for approximating Hessians and kernel functions, respectively. Recall that a Givens rotation matrix [21] is a sparse matrix with one angle as its parameter that rotates two dimensions by this angle. If we assume for the moment that $d = 2^k$, $k \in \mathbb{N}^*$, then we consider $k$ rotation matrices denoted $\mathcal{O}_1, \ldots \mathcal{O}_k$ where for any $i \in \{1, \ldots, k\}$, $\mathcal{O}_i$ contains $d/2$ independent rotations, *i.e.* is the product of $d/2$ independent Givens rotations. Givens rotations are arranged in a butterfly architecture that provides for a minimal number of rotations so that all coordinates can interact with one another in the rotation defined by $\mathcal{R}_d$. For illustration, consider

the case $d = 4$, where the rotation matrix is fully described using $4 - 1$ parameters $\nu_1, \nu_2, \nu_3 \in \mathbb{R}$ by $\mathcal{R}_4 = \mathcal{O}_1 \mathcal{O}_2$ with

$$\mathcal{O}_1 \mathcal{O}_2 = \begin{bmatrix} c_1 & -s_1 & 0 & 0 \\ s_1 & c_1 & 0 & 0 \\ 0 & 0 & c_3 & -s_3 \\ 0 & 0 & s_3 & c_3 \end{bmatrix} \begin{bmatrix} c_2 & 0 & -s_2 & 0 \\ 0 & c_2 & 0 & -s_2 \\ s_2 & 0 & c_2 & 0 \\ 0 & s_2 & 0 & c_2 \end{bmatrix} = \begin{bmatrix} c_1 c_2 & -s_1 c_2 & -c_1 s_2 & s_1 s_2 \\ s_1 c_2 & c_1 c_2 & -s_1 s_2 & -c_1 s_s \\ c_3 s_2 & -s_3 s_2 & c_3 c_2 & -s_3 c_s \\ s_3 s_2 & c_3 s_2 & s_3 c_2 & c_3 c_2 \end{bmatrix},$$

where $c_i = \cos(\nu_i)$ and $s_i = \sin(\nu_i)$. We provide a precise recursive definition of $\mathcal{R}_d$ in Appendix B where we also describe the case where $d$ is not a power of two. In general, we have a computational complexity of $\mathcal{O}(d \log d)$, due to the fact that $\mathcal{R}_d$ is a product of $\mathcal{O}(\log d)$ matrices each requiring $\mathcal{O}(d)$ operations. Moreover, note that $\mathcal{R}_d$ is parametrized by $d - 1$ parameters $(\nu_i)_{i \in \{1 \ldots d-1\}}$ and each $\mathcal{O}_i$ can be implemented as a sparse matrix, which implies a memory complexity of $\mathcal{O}(d)$. Furthermore, since $\mathcal{O}_i$ is orthonormal, we have $\mathcal{O}_i^{-1} = \mathcal{O}_i^\top$ and $|\det \mathcal{O}_i| = 1$.

To construct an expressive variational distribution, we consider as a base distribution $q_0$ the proposed copula-like density $c_\theta$. We then apply the transformations $\mathcal{T}_1 = \mathcal{H}$ and $\mathcal{T}_2 = \mathcal{G}$. The operator $\mathcal{G}$ in (5) is defined via quantile functions of densities $f_1, \ldots, f_d$, for which we choose Gaussian densities with parameter $\phi_f = (\mu_1, \ldots, \mu_d, \sigma_1^2, \ldots, \sigma_d^2) \in \mathbb{R}^d \times \mathbb{R}_+^d$. As a final transformation, we apply the volume-preserving operator

$$\mathcal{T}_3 : x \mapsto \mathcal{O}_1 \cdots \mathcal{O}_{\log d} x \tag{9}$$

that has parameter $\phi_\mathcal{R} = (\nu_1, \ldots, \nu_{d-1}) \in \mathbb{R}^{d-1}$. Altogether, the parameter for the marginal-like densities that we optimize over is $\phi = (\phi_f, \phi_\mathcal{R})$ and simulation from the variational density boils down to the following algorithm.

---

**Algorithm 1** Sampling from the rotated copula-like density.

---

1: Sample $(V_1, \ldots, V_d) \sim c_\theta$ using Proposition 1.
2: Set $U = \mathcal{H}(V)$ where $\mathcal{H}$ is defined in (7).
3: Set $X' = \mathcal{G}(U)$, where $\mathcal{G}$ is defined in (5).
4: Set $X = \mathcal{T}_3$, where $\mathcal{T}_3$ is defined in (9).

---

Note that we apply the rotations after we have transformed samples from the hypercube into $\mathbb{R}^d$, as the hypercube is not closed under Givens rotations. The variational density can then be evaluated using the normalizing flow formula (3). We optimize the variational lower bound $\mathcal{L}$ in (2) using reparametrization gradients, proposed by [35, 60, 63], but with an implicit reparametrization, cf. [14], for Dirichlet and Beta distributions. Such reparametrized gradients for Dirichlet and Beta distributions are readily available for instance in tensorflow probability [9]. Using Monte Carlo samples of unbiased gradient estimates, one can optimize the variational bound using some version of stochastic gradient descent. A more formal description is given in Appendix C.

We would like to remark that such sparse rotations can be similarly applied to proper copulas. While there is no additional flexibility by rotating a full-rank Gaussian copula, applying such rotations to a Gaussian copula with a low-rank correlation yields a Gaussian distribution with a more flexible covariance structure if combined with Gaussian marginals. In our experiments, we therefore also compare variational families constructed by sampling $(V_1, \ldots, V_d)$ from an independence copula in step 1 in Algorithm 1, *i.e.* $V_i$ are independent and uniformly distributed on $[0, 1]$ for any $i \in \{1, \ldots, d\}$, which results approximately in a Gaussian variational distribution if the effect of the transformation $\mathcal{H}$ is neglected. However, a more thorough analysis of such families is left for future work. Similarly, transformations different from the sparse rotations in step 4 in Algorithm 1 can be used in combination with a copula-like base density. Whilst we include a comparison with a simple Inverse Autoregressive Flow [34] in our experiments, a more exhaustive study of non-linear transformations is beyond the scope of this work.

## 5 Related Work

Conceptually, our work is closely related to [66, 23]. It differs from [66] in that it can be applied in high dimensions without having to search first for the most correlated variables using for instance a sequential tree selection algorithm [11]. The approach in [23] considered a Gaussian dependence structure, but has only been considered in low-dimensional settings. On a more computational side,

our approach is related to variational inference with normalizing flows [59, 34, 65, 43, 3]. In contrast to these works that introduce a parameter-free base distribution commonly in $\mathbb{R}^d$ as the latent state space, we also optimize over the parameters of the base distribution which is supported on the hypercube instead, although distributions supported for instance on the hypersphere as a state space have been considered in [7]. Moreover, such approaches have been often used in the context of generative models using Variational Auto-Encoders (VAEs) [35], yet it is in principle possible to apply the proposed variational copula-like inference in an amortized fashion for VAEs.

A somewhat similar copula-like construction in the context of importance sampling has been proposed in [8]. However, sampling from this density requires a rejection step to ensure support on the hypercube, which would make optimization of the variational bound less straightforward. Lastly, [30] proposed a method to approximate copulas using mixture distributions, but these approximations have not been analysed neither in high dimensions nor in the context of variational inference.

# 6 Experiments

## 6.1 Bayesian Logistic Regression

Consider the target distribution $\pi$ on $(\mathbb{R}^d, \mathcal{B}(\mathbb{R}^d))$ arising as the posterior of a $d$-dimensional logistic regression, assuming a Normal prior $\pi_0 = \mathcal{N}(0, \tau^{-1}I)$, $\tau = 0.01$, and likelihood function $L(y^i|x) = f(y^i x^\top \mathrm{a}^i)$, $f(z) = 1/(1 + \mathrm{e}^{-z})$ with $n$ observations $y^i \in \{-1, 1\}$ and fixed covariates $\mathrm{a}^i \in \mathbb{R}^d$ for $i \in \{1, \dots n\}$. We analyse a previously considered synthetic dataset where the posterior distribution is non-Gaussian, yet it can be well approximated with our copula-like construction. Concretely, we consider the synthetic dataset with $d = 2$ as in [50], Section 8.4 and [32] by generating 30 covariates $\mathrm{a} \in \mathbb{R}^2$ from a Gaussian $\mathcal{N}((1, 5)^\top, I)$ for instances in the first class, while we generate 30 covariates from $\mathcal{N}((-5, 1)^\top, 1.1^2 I)$ for instances in the second class. Samples from the target distribution using a Hamiltonian Monte Carlo (HMC) sampler [12, 51] are shown in Figure 1a and one observes non-Gaussian marginals that are positively correlated with heavy right tails. Using a Gaussian variational approximation with either independent marginals or a full covariance matrix as shown in Figure 1b does not adequately approximate the target distribution. Our copula-like construction is able to approximate the target more closely, both without any rotations (Figure 1c) and with a rotation of the marginals (Figure 1d). This is also supported by the ELBO obtained for the different variational families given in Table 1.

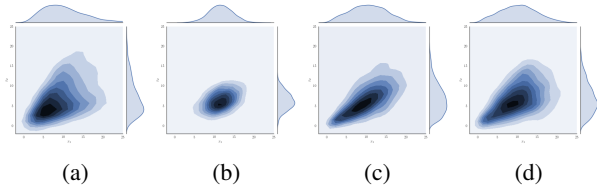

(a)      (b)      (c)      (d)

Table 1: Comparison of the ELBO between different variational families for the logistic regression experiment.

| Variational family | ELBO |
|---|---|
| Mean-field Gaussian | -3.42 |
| Full-covariance Gaussian | -2.97 |
| Copula-like without rotations | -2.30 |
| Copula-like with rotations | -2.19 |

Figure 1: Target density for logistic regression using a HMC sampler in 1a with different variational approximations: Gaussian variational approximation with a full covariance matrix in 1b, copula-like variational approximation without any rotation in 1c and copula-like variational approximation with a rotation in 1d.

## 6.2 Centred Horseshoe Priors

We illustrate our approach in a hierarchical Bayesian model that posits a priori a strong coupling of the latent parameters. As an example, we consider a Horseshoe prior [6] that has been considered in the variational Gaussian copula framework in [23]. To be more specific, consider the generative model $y|\lambda \sim \mathcal{N}(0, \lambda)$, with $\lambda \sim \mathcal{C}^+(0, 1)$, where $\mathcal{C}^+$ is a half-Cauchy distribution, *i.e.* $X \sim \mathcal{C}^+(0, b)$ has the density $p(x) \propto 1_{\mathbb{R}_+}(x)/(x^2 + b^2)$. Note that we can represent a half-Cauchy distribution with Inverse Gamma and Gamma distributions using $X \sim \mathcal{C}^+(0, b) \iff X^2|Y \sim \mathcal{IG}(1/2, 1/Y); Y \sim \mathcal{IG}(1/2, 1/b^2)$, see [52], with a rate parametrisation of the inverse gamma density $p(x) \propto 1_{\mathbb{R}_+}(x)x^{a-1}e^{-b/x}$ for $X \sim \mathcal{IG}(a, b)$. We revisit the toy model in [23] fixing $y = 0.01$.

The model thus writes in a centred form as $\eta \sim \mathcal{G}(1/2, 1)$ and $\lambda|\eta \sim \mathcal{IG}(1/2, \eta)$. Following [23], we consider the posterior density on $\mathbb{R}^2$ of the log-transformed variables $(x_1, x_2) = (\log \eta_1, \log \lambda_1)$. In Figure 2, we show the approximate posterior distribution using a Gaussian family (2b) and a copula-like family (2c), together with samples from a HMC sampler (2a). A copula-like density yields a higher ELBO, see Table 2. The experiments in [23] have shown that a Gaussian copula with a non-parametric mixture model fits the marginals more closely. To illustrate that it is possible to arrive at a more flexible variational family by using a mixture of copula-like densities, we have used a mixture of 3 copula-like densities in Figure 2d. Note that it is possible to accommodate multi-modal marginals using a Gaussian quantile transformation with a copula-like density. Eventually, the flexibility of the variational approximation can be increased using different complementary work. For instance, one could use the new density within a semi-implicit variational framework [69] whose parameters are the output of a neural network conditional on some latent mixing variable.

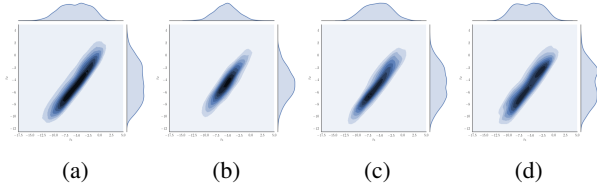

(a)       (b)       (c)       (d)

Table 2: Comparison of the ELBO between different variational families for the centred horseshoe model.

| Variational family | ELBO |
|---|---|
| Mean-field Gaussian | -1.24 |
| Full-covariance Gaussian | -0.04 |
| Copula-like | 0.04 |
| 3-mixture copula-like | 0.08 |

Figure 2: Target density for the horseshoe model using a HMC sampler in 2a with different variational approximations: Gaussian variational approximation with a full covariance matrix in 2b, copula-like variational approximation including a rotation in 2c and a mixture of three copula-like densities with a one rotation and marginal-like density in 2d.

## 6.3 Bayesian Neural Networks with Normal Priors

We consider an $L$-hidden layer fully-connected neural network where each layer $l$, $1 \leqslant l \leqslant L + 1$ has width $d_l$ and is parametrised by a weight matrix $W^l \in \mathbb{R}^{d_{l-1} \times d_l}$ and bias vector $b^l \in \mathbb{R}^{d_l}$. Let $h^1 \in \mathbb{R}^{d_0}$ denote the input to the network and $f$ be a point-wise non-linearity such as the ReLU function $f(a) = \max\{0, a\}$ and define the activations $a^l \in \mathbb{R}^{d_l}$ by $a^{l+1} = \sum_i h_i^l W_{i\cdot}^l + b^l$ for $l \geqslant 1$, and the post-activations as $h^l = f(a^l)$ for $l \geqslant 2$. We consider a regression likelihood function $L(\cdot|a^{L+2}, \sigma) = \mathcal{N}(a^{L+2}, \exp(0.5\sigma))$, and denote the concatenation of all parameters $W$, $b$ and $\sigma$ as $x$. We assume independent Normal priors for the entries of the weight matrix and bias vector with mean 0 and variance $\sigma_0^2$. Furthermore, we assume that $\log \sigma \sim \mathcal{N}(0, 16)$. Inference with the proposed variational family is applied on commonly considered UCI regression datasets, repeating the experimental set-up used in [15]. In particular, we use neural networks with ReLU activation functions and one hidden layer of size 50 for all datasets with the exception of the protein dataset that utilizes a hidden layer of size 100. We choose the hyper-parameter $\sigma_0^2 \in \{0.01, 0.1, 1., 10., 100.\}$ that performed best on a validation dataset in terms of its predictive log-likelihood. Optimization was performed using Adam [33] with a learning rate of 0.002. We compare the predictive performance of a copula-like density $c_\theta$ and an independent copula as a base distribution in step 1 of Algorithm 1 and we apply different transformations $\mathscr{T}_3$ in step 4 of Algorithm 1: a) the proposed sparse rotation defined in (9); b) $\mathscr{T}_3 = \mathrm{Id}$; c) an affine autoregressive transformation $\mathscr{T}_3(x) = \{x - f_\mu(x)\}\exp(-f_\alpha(x))$, see [34], also known as an inverse autoregressive flow (IAF). Here $f_\mu$ and $f_\alpha$ are autoregressive neural networks parametrized by $\mu$ and $\alpha$ satisfying $\frac{\partial f_\mu(x)_i}{\partial x_j} = \frac{\partial f_\alpha(x)_i}{\partial x_j} = 0$ for $i \leqslant j$ and which can be computed in a single forward pass by properly masking the weights in the neural networks [17]. In our experiments, we use a one-hidden layer fully-connected network with width $d_1^{\mathrm{IAF}} = 50$ for $f_\mu$ and $f_\alpha$. Note that for a $d$-dimensional target density, the size of the weight matrices are of order $d \cdot d_1^{\mathrm{IAF}}$, implying a higher complexity compared to the sparse rotation. We also compare the predictions against Bayes-by-Backprop [5] using a mean-field model, with the results as reported in [47] for a mean-field Bayes-by-Backprop and low-rank Gaussian approximation proposed therein called SLANG. Furthermore, we also report the results for Dropout inference [15]. The test root mean-squared errors are given in Table 3 and

Table 3: Variational approximations with transformations and different base distributions. Test root mean-squared error for UCI regression datasets. Standard errors in parenthesis.

|  | Copula-like with rotation | Independent copula with rotation | Copula-like with IAF | Independent copula with IAF |
|---|---|---|---|---|
| Boston | 3.43 (0.22) | 3.51 (0.30) | 3.21 (0.27) | 3.61 (0.28) |
| Concrete | 5.76 (0.14) | 6.00 (0.13) | 5.41 (0.10) | 5.82 (0.11) |
| Energy | 0.55 (0.01) | 2.28 (0.11) | 0.53 (0.02) | 1.30 (0.10) |
| Kin8nm | **0.08 (0.00)** | **0.08 (0.00)** | **0.08 (0.00)** | **0.08 (0.00)** |
| Naval | **0.00 (0.00)** | **0.00 (0.00)** | **0.00 (0.00)** | **0.00 (0.00)** |
| Power | **4.02 (0.04)** | 4.19 (0.04) | 4.05 (0.04) | 4.15 (0.04) |
| Wine | 0.64 (0.01) | 0.64 (0.01) | 0.64 ( 0.01) | 0.64 (0.01) |
| Yacht | 1.35 (0.08) | 1.38 (0.12) | **0.96 (0.06)** | 1.25 (0.09) |
| Protein | **4.20 (0.01)** | 4.51 (0.04) | 4.31 (0.01) | 4.51 (0.03) |

Table 4: Copula-like variational approximation without rotations and benchmark results. Test root mean-squared error for UCI regression datasets. Standard errors in parenthesis.

|  | Copula-like without rotation | Bayes-by-Backprop results from [47] | SLANG results from [47] | Dropout results from [47] |
|---|---|---|---|---|
| Boston | 3.22 (0.25) | 3.43 (0.20) | 3.21 (0.19) | **2.97 (0.19)** |
| Concrete | 5.64 (0.14) | 6.16 (0.13) | 5.58 (0.12) | **5.23 (0.12)** |
| Energy | **0.52 (0.02)** | 0.97 (0.09) | 0.64 (0.04) | 1.66 (0.04) |
| Kin8nm | **0.08 (0.00)** | **0.08 (0.00)** | **0.08 (0.00)** | 0.10 (0.01) |
| Naval | **0.00 (0.00)** | **0.00 (0.00)** | **0.00 (0.00)** | 0.01 (0.01) |
| Power | 4.05 (0.04) | 4.21 (0.03) | 4.16 (0.04) | **4.02 (0.04)** |
| Wine | 0.65 (0.01) | 0.64 (0.01) | 0.65 ( 0.01) | **0.62 (0.01)** |
| Yacht | 1.23 (0.08) | 1.13 (0.06) | 1.08 (0.09) | 1.11 (0.09) |
| Protein | 4.31 (0.02) | NA | NA | 4.27 (0.01) |

Table 4; the predictive test log-likelihood can be find in the Appendix E in Table 6 and Table 7. We can observe from Table 3 and Table 6 that using a copula-like base distribution instead of an independent copula improves the predictive performance, using either rotations or IAF as the final transformation. The same tables also indicate that for a given base distribution, the IAF tends to outperform the sparse rotations slightly. Table 4 and Table 7 suggest that copula-like densities without any transformation in the last step can be a competitive alternative to a benchmark mean-field or Gaussian approximation. Dropout tends to perform slightly better. However, note that Dropout with a Normal prior and a variational mixture distribution that includes a Dirac delta function as one component gives rise to a different objective, since the prior is not absolutely continuous with respect to the approximate posterior, see [25].

## 6.4 Bayesian Neural Networks with Structured Priors

We illustrate our approach on a larger Bayesian neural network. To induce sparsity for the weights in the network, we consider a (regularised) Horseshoe prior [56] that has also been used increasingly as an alternative prior in Bayesian neural network to allow for sparse variational approximations, see [41, 19] for mean-field models and [20] for a structured Gaussian approximation. We consider again an $L$-hidden layer fully-connected neural network where we assume that the weight matrix $W^l \in \mathbb{R}^{d_{l-1} \times d_l}$ for any $l \in \{1, \ldots, L+1\}$ and any $i \in \{1, \ldots, d_{l-1}\}$ satisfies a priori

$$W_{i\cdot}^l | \lambda_i^l, \tau^l, c \sim \mathcal{N}(0, (\tau^l \tilde{\lambda}_i^l)^2 I) \propto \mathcal{N}(0, (\tau^l \lambda_i^l))^2 I) \mathcal{N}(0, c^2), \tag{10}$$

where $(\tilde{\lambda}_i^l)^2 = c^2 (\lambda_i^l)^2 / (c^2 + \tau^2 (\lambda_i^l)^2)$, $\lambda_i^l \sim \mathcal{C}^+(0,1)$, $\tau_i^l \sim \mathcal{C}^+(0, b_\tau)$ and $c^2 \sim \mathcal{IG}(\frac{\nu}{2}, \nu \frac{s^2}{2})$ for some hyper-parameters $b_\tau, \nu, s^2 > 0$. The vector $W_{i\cdot}^{(l)}$ represents all weights that interact with the $i$-th input neuron. The first Normal factor in (10) is a standard Horseshoe prior with a per layer global parameter $\tau^l$ that adapts to the overall sparsity in layer $l$ and shrinks all weights in this layer to zero, due to the fact that $\mathcal{C}^+(0, b_\tau)$ allows for substantial mass near zero. The local shrinkage

Table 5: MNIST prediction errors.

| Variational approximation with Horseshoe prior and size $200 \times 200$ | Error Rate |
|---|---|
| Copula-like with rotations | 1.70 % |
| Copula-like without rotations | 1.78 % |
| Copula-like with IAF | 2.04 % |
| Independent copula with IAF | 2.88 % |
| Independent copula with rotations | 2.90 % |
| Mean-field Gaussian | 3.82 % |
| Copula-like without rotations and $\delta_i = 0.99$ for all $i \in \{1, \ldots, d\}$ | 5.70 % |

parameter $\lambda_i^l$ allow for signals in the $i$-th input neuron because $\mathcal{C}^+(0,1)$ is heavy-tailed. However, this can leave large weights un-shrunk, and the second Normal factor in (10) induces a Student-$t_\nu(0, s^2)$ regularisation for weights far from zero, see [56] for details. We can rewrite the model in a non-centred form [55], where the latent parameters are a priori independent, see also [41, 27, 19, 20] for similar variational approximations. We write the model as $\eta_i^l \sim \mathcal{G}(1/2, 1)$, $\hat{\lambda}_i^l \sim \mathcal{IG}(1/2, 1)$, $\kappa^l \sim \mathcal{G}(1/2, 1/b_\tau^2)$, $\hat{\tau}^l \sim \mathcal{IG}(1/2, 1)$, $\beta_i^l \sim \mathcal{N}(0, I)$, $W_{i\cdot}^l = \tau^l \tilde{\lambda}_i^l \beta_i^l$, $\tau^l = \sqrt{\hat{\tau}^l \kappa^l}$, $\lambda_i^l = \sqrt{\hat{\lambda}_i^l \eta_i^l}$ and $(\tilde{\lambda}_i^l)^2 = c^2 (\lambda_i^l)^2 / (c^2 + (\tau^l)^2 (\lambda_i^l)^2)$. The target density is the posterior of these variables, after applying a log-transformation if their prior is an (inverse) Gamma law.

We performed classification on MNIST using a 2-hidden layer fully-connected network where the hidden layers are of size 200 each. Further details about the algorithmic details are given in Appendix D. Prediction errors for the variational families as considered in the preceding experiments are given in Table 5. We again find that a copula-like density outperforms the independent copula. Using a copula-like density without the rotation also performs competitively as long as one uses a balanced amount of its antithetic component via the transformation $\mathscr{H}$ with parameter $\delta$; ignoring the transformation $\mathscr{H}$ or setting $\delta_i = 0.99$ for all $i \in \{1, \ldots, d\}$ can limit the representative power of the variational family and can result in high predictive errors. The neural network function for the IAF considered here has two hidden layers of size $100 \times 100$. It can be seen that applying the rotations can be beneficial compared to the IAF for the copula-like density, whereas the two transformations perform similarly for the independent base distribution. We expect that more ad-hoc tricks can be used to adjust the rotations to some computational budget. For instance, one could include additional rotations for a group of latent variables such as those within one layer. Conversely, one could consider the series of sparse rotations $\mathcal{O}_1, \cdots, \mathcal{O}_k$, but with $2^k < d$, thereby allowing for rotations of the more adjacent latent variables only.

Our experiment illustrates that the proposed approach can be used in high-dimensional structured Bayesian models without having to specify more model-specific dependency assumptions in the variatonal family. The prediction errors are in line with current work for fully connected networks using a Gaussian variational family with Normal priors, cf. [47]. Better predictive performance for a fully connected Bayesian network has been reported in [37] that use RealNVP [10] as a normalising flows in a large network that is reparametrised using a weight normalization [61]. It becomes scalable by opting to consider only variational inference over the Euclidean norm of $W_{i\cdot}^l$ and performing point estimation for the direction of the weight vector $W_{i\cdot}^l / \|W_{i\cdot}^l\|_2$. Such a parametrisation does not allow for a flexible dependence structure of the weights within one layer; and such a model architecture should be complementary to the proposed variational family in this work.

## 7  Conclusion

We have addressed the challenging problem of constructing a family of distributions that allows for some flexibility in its dependence structure, whilst also having a reasonable computational complexity. It has been shown experimentally that it can constitute a useful replacement of a Gaussian approximation without requiring many algorithmic changes.

## Acknowledgements

Alain Durmus acknowledges support from Chaire BayeScale "P. Laffitte" and from Polish National Science Center grant: NCN UMO-2018/31/B/ST1/0025. This research has been partly financed by the Alan Turing Institute under the EPSRC grant EP/N510129/1. The authors acknowledge the use of the UCL Myriad High Throughput Computing Facility (Myriad@UCL), and associated support services, in the completion of this work.

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
