[Supplementary Material]

# Appendices

## A Proof of Proposition 1

*Proof.* Let $f : \mathbb{R}^d \to \mathbb{R}_+$ be a positive and bounded function. We have by definition, using the expression of the density of the Dirichlet and Beta distributions, see [13], and setting $u_d = 1 - \sum_{i=1}^{d-1} u_i$,

$$
\mathbb{E}\left[f(V_1,\ldots,V_n)\right] = \frac{\boldsymbol{\Gamma}(\alpha^\star)}{\mathrm{B}(a,b)} \int_{[0,1]^d} f\left\{gu_1 / \max_{j\in\{1,\ldots,d\}} u_j, \ldots, gu_d / \max_{j\in\{1,\ldots,d\}} u_j\right\}
$$

$$
\times\, g^{a-1}(1-g)^{b-1} \left\{\prod_{\ell=1}^{d} \frac{u_\ell^{\alpha_\ell-1}}{\boldsymbol{\Gamma}(\alpha_\ell)}\right\} \mathrm{Leb}(g,u_1,\ldots,u_{d-1})
$$

$$
= \sum_{k=1}^{d} \frac{\boldsymbol{\Gamma}(\alpha^\star)}{\mathrm{B}(a,b)} A_k \,, \tag{11}
$$

where

$$
A_k = \int_{[0,1]^d} \mathbb{1}\left\{u_k = \max_{j\in\{1,\ldots,d\}} u_j\right\} f\left\{gu_1/u_k,\ldots,gu_d/u_k\right\}
$$

$$
\times\, g^{a-1}(1-g)^{b-1} \left\{\prod_{\ell=1}^{d} \frac{u_\ell^{\alpha_\ell-1}}{\boldsymbol{\Gamma}(\alpha_\ell)}\right\} \mathrm{Leb}(g,u_1,\ldots,u_{d-1}) \,. \tag{12}
$$

Then by symmetry, without loss of generality, we only need to consider $A_1$. Using the change of variable, $(g, u_1, u_2, \ldots, u_{d-1}) \mapsto (g, u_1, gu_2/u_1, \ldots, gu_{d-1}/u_1)$, which is a $\mathrm{C}^1$-diffeomorphism from $\Delta_1 = \{(g, u_1, \ldots, u_{d-1}) \in [0,1]^d : u_1 = \max_{j\in\{1,\ldots,d\}} u_j\}$ to $\tilde{\Delta}_1 = \{(g, u_1, w_2, \ldots, w_{d-1}) \in [0,1]^d : \max_{j\in\{2,\ldots,d-1\}} w_j \leqslant g, g/u_1 - g - \sum_{j=2}^{d-1} w_j \leqslant g\}$, we get that

$$
A_1 = \int_{\Delta_1} f\left\{g,\ldots,gu_d/u_1\right\} g^{a-1}(1-g)^{b-1} \left\{\prod_{\ell=1}^{d} \frac{u_\ell^{\alpha_\ell-1}}{\boldsymbol{\Gamma}(\alpha_\ell)}\right\} \mathrm{Leb}(g,u_1,u_2,\ldots,u_{d-1})
$$

$$
= \int_{\tilde{\Delta}_1} f\left\{g, w_2, \ldots, w_{d-1}, g/u_1 - g - \sum_{i=2}^{d-1} w_i\right\} g^{a-1}(1-g)^{b-1}
$$

$$
\times \left\{\prod_{\ell=2}^{d-2} \frac{(u_1 w_\ell/g)^{\alpha_\ell-1}}{\boldsymbol{\Gamma}(\alpha_\ell)}\right\} \frac{u_1^{\alpha_1-1}}{\boldsymbol{\Gamma}(\alpha_1)} \frac{(1 - u_1 - \sum_{i=2}^{d-1} u_1 w_i/g)^{\alpha_d-1}}{\boldsymbol{\Gamma}(\alpha_d)} \frac{g^{d-2}}{u_1^{d-2}} \mathrm{Leb}(g, u_1, w_2, \ldots, w_{d-1})
$$

$$
= \int_{\tilde{\Delta}_1} f\left\{g, w_2, \ldots, w_{d-1}, g/u_1 - g - \sum_{i=2}^{d-1} w_i\right\} g^{a-1}(1-g)^{b-1}
$$

$$
\times \left\{\prod_{\ell=2}^{d-2} \frac{w_\ell^{\alpha_\ell-1}}{\boldsymbol{\Gamma}(\alpha_\ell)}\right\} \frac{u_1^{\alpha^\star-2}}{\boldsymbol{\Gamma}(\alpha_1)} \frac{(g/u_1 - g - \sum_{i=2}^{d-1} w_i)^{\alpha_d}}{\boldsymbol{\Gamma}(\alpha_d)} g^{-\alpha^\star+\alpha_1+1} \mathrm{Leb}(g, u_1, w_2, \ldots, w_{d-1})
$$

$$
= \int_{\tilde{\Delta}_1} f\left\{g, w_2, \ldots, w_{d-1}, g/u_1 - g - \sum_{i=2}^{d-1} w_i\right\} g^{a-1}(1-g)^{b-1}
$$

$$
\times \left\{\prod_{\ell=2}^{d-2} \frac{w_\ell^{\alpha_\ell-1}}{\boldsymbol{\Gamma}(\alpha_\ell)}\right\} \frac{g^{\alpha_1-1}}{\boldsymbol{\Gamma}(\alpha_1)} \frac{(g/u_1 - g - \sum_{i=2}^{d-1} w_i)^{\alpha_d-1}}{\boldsymbol{\Gamma}(\alpha_d)} (u_1/g)^{\alpha^\star-2} \mathrm{Leb}(g, u_1, w_2, \ldots, w_{d-1}) \,.
$$

Now using the change of variable $(g, u_1, w_2, \ldots, w_{d-1}) \mapsto (g, g/u_1 - \sum_{i=2}^{d-1} w_i, w_2, \ldots, w_{d-1}) = (g, w_d, \ldots, w_{d-1})$, which is a $\mathrm{C}^1$-diffeomorphism from $\tilde{\Delta}_1$ to

$$
\bar{\Delta}_1 = \{(g, w_d, w_2, \ldots, w_{d-1}) : \max_{j\in\{1,\ldots,d\}} w_j \leqslant g\} \,,
$$

we obtain since $g/u_1 = g + \sum_{j=2}^{d} w_j$ that

$$A_1 = \int_{\bar{\Delta}_1} f(g, w_2, \ldots, w_{d-1}, w_d)) g^{a-1}(1-g)^{b-1}$$

$$\times \left\{ \prod_{\ell=2}^{d} \frac{w_\ell^{\alpha_\ell - 1}}{\mathbf{\Gamma}(\alpha_\ell)} \right\} \frac{g^{\alpha_1}}{\mathbf{\Gamma}(\alpha_1)} \left\{ g + \sum_{j=1}^{d-1} w_j \right\}^{-\alpha^\star} \mathrm{Leb}(g, w_1, w_2, \ldots, w_{d-1}).$$

Combining this result, (11) and (12) completes the proof. $\qquad\square$

## B    Butterfly rotation matrices

Suppose $d = 2^k$ for some $k \in \mathbb{N}$ and let $c_i = \cos \nu_i$ and $s_i = \sin \nu_i$. For $d = 1$, define $\mathcal{R}_1 = [1]$. Assume $\mathcal{R}_d$ has been defined. Then define

$$\mathcal{R}_{2d} = \begin{bmatrix} \mathcal{R}_d c_d & -\mathcal{R}_d s_d \\ \tilde{\mathcal{R}}_d s_d & \tilde{\mathcal{R}}_d c_d \end{bmatrix},$$

where $\tilde{\mathcal{R}}_d$ has the same form as $\mathcal{R}_d$ except that the $c_i$ and $s_i$ indices are all increased by $d$. So for instance

$$\mathcal{R}_2 = \begin{bmatrix} c_1 & -s_1 \\ s_1 & c_1 \end{bmatrix} \quad , \quad \tilde{\mathcal{R}}_2 = \begin{bmatrix} c_3 & -s_3 \\ s_3 & c_3 \end{bmatrix}.$$

Suppose now that $d$ is not a power of 2 and let $k = \lceil \log d \rceil$. We construct $\mathcal{R}_d$ as a product of $k$ factors $\mathcal{O}_1 \cdots \mathcal{O}_k$ as used in the construction of $\mathcal{R}_{2^k}$. For any $i \in \{1, \ldots k\}$, we then delete from $\mathcal{O}_i$ the last $2^k - d$ rows and columns. Then for every $c_i$ in the remaining $d \times d$ matrix that is in the same column as a deleted $s_i$ is replaced by 1. As an example, for $d = 5$, we have

$$\mathcal{R}_5 = \begin{bmatrix} c_1 & -s_1 & 0 & 0 & 0 \\ s_1 & c_1 & 0 & 0 & 0 \\ 0 & 0 & c_3 & -s_3 & 0 \\ 0 & 0 & s_3 & c_3 & 0 \\ 0 & 0 & 0 & 0 & 1 \end{bmatrix} \begin{bmatrix} c_2 & 0 & -s_2 & 0 & 0 \\ 0 & c_2 & 0 & -s_2 & 0 \\ s_2 & 0 & c_2 & 0 & 0 \\ 0 & s_2 & 0 & c_2 & 0 \\ 0 & 0 & 0 & 0 & 1 \end{bmatrix} \begin{bmatrix} c_4 & 0 & 0 & 0 & -s_4 \\ 0 & 1 & 0 & 0 & 0 \\ 0 & 0 & 1 & 0 & 0 \\ 0 & 0 & 0 & 1 & 0 \\ s_4 & 0 & 0 & 0 & c_4 \end{bmatrix}.$$

## C    Optimization of the variational bound

Recall that for independent random variables $Z_i \sim \mathcal{G}(\alpha_i, 1)$, for $i \in \{1, \ldots d\}$, we have $\left( \frac{Z_1}{\sum_{j=1}^{d} Z_j}, \cdots \frac{Z_d}{\sum_{j=1}^{d} Z_j} \right) \sim \mathrm{Dirichlet}(\alpha_1, \ldots, \alpha_d)$, cf. [13]. Similarly, for independent random variables $Z_{d+1} \sim \mathcal{G}(a, 1)$ and $Z_{d+2} \sim \mathcal{G}(b, 1)$, it holds that $\frac{Z_{d+1}}{Z_{d+1} + Z_{d+2}} \sim \mathrm{Beta}(a, b)$. Recall that the parameter of the rotated variational family is $\xi = (\theta, \phi, \delta)$, where $\theta$ is the parameter of the copula-like base density, whereas $\phi = (\phi_f, \phi_{\mathcal{R}})$ denotes the parameters of the quantile transformation and the rotation, respectively. Furthermore, the parameter $\delta$ of the transformation $\mathcal{H}$ is kept fix. Using Proposition 1 and Algorithm 1 for some fixed $\delta$, we can construct a function $(z, \phi) \mapsto f_{\phi, \delta}(z)$, $z = (z_1, \ldots z_{d+2})$, that is almost everywhere continuously differentiable such that $f_{\phi, \delta}(Z_1, \ldots Z_{d+2}) \sim q_\xi$, where $q_\xi$ is the density of the proposed variational family with parameter $\xi = (\theta, \phi, \delta)$, that is the variational density $q_\xi$ is the pushforward density of independent Gamma densities with parameter $\theta$ through the transport map $f_{\phi, \delta}$. Differentiability with respect to $\phi_f$ can be achieved by a continuous numerical approximation for the quantile function of a standard Gaussian and applying appropriate (re)normalisation. Furthermore, there exists an invertible standardization function $\mathcal{S}_\theta$ with $(z, \theta) \mapsto \mathcal{S}_\theta(z) = (\mathbb{P}(Z_1 \leqslant z_1), \ldots, \mathbb{P}(Z_{d+2} \leqslant z_{d+2}))$ continuously differentiable such that $\mathcal{S}_\theta^{-1}(H)$ is equal to $(Z_1, \ldots Z_{d+2})$ in distribution, where $H$ is a $(d+2)$-dimensional vector of iid random variables with uniform marginals on $[0, 1]$. In particular, the distribution of $H$ does not depend on $\xi$. The cumulative distribution function of $Z_1$ say at the point $z_1$ is the regularised incomplete Gamma function $\gamma(z_1, \alpha_1)$ that lacks an analytical expression though. However, one can apply automatic differentiation to a numerical method that approximates $\gamma(z_1, \alpha_1)$ yielding an approximation of $\frac{\partial \gamma(z_1, \alpha_1)}{\partial \alpha_1}$. Let us define

$$l(z, \phi, \delta) = \frac{\log L(y^{1:n} | f_{\phi, \delta}(z)) + \log \pi_0(f_{\phi, \delta}(z))}{\log q_\xi(f_{\phi, \delta}(z))}.$$

Then $\mathcal{L}(\xi) = \mathbb{E}\left[l(Z, \phi, \delta)\right] = \mathbb{E}\left[l(\mathcal{S}_\theta^{-1}(H), \phi, \delta)\right]$, where in the first expectation, the law of the random variable $Z$ depends on $\theta$. For a differentiable function $g \colon \mathbb{R}^n \to \mathbb{R}^m$, we denote by $\nabla_x g(x)$ the Jacobian of $g$, that is $\nabla_x g(x)_{ij} = \frac{\partial g_i(x)}{\partial x_j}$. Following the arguments in [14], we obtain for the Jacobian of the variational bound

$$
\begin{aligned}
\nabla_{\theta,\phi}\mathcal{L}(\xi) &= \mathbb{E}\left[\nabla_{\theta,\phi}l(\mathcal{S}_\theta^{-1}(H), \phi, \delta)\right] \\
&= \mathbb{E}\left[\nabla_z l(\mathcal{S}_\theta^{-1}(H), \phi, \delta)\nabla_{\theta,\phi}\mathcal{S}_\theta^{-1}(H) + \nabla_{\theta,\phi}l(\mathcal{S}_\theta^{-1}(H), \phi, \delta)\right] \\
&= \mathbb{E}\left[\nabla_z l(Z, \phi, \delta)\nabla_{\theta,\phi}Z + \nabla_{\theta,\phi}l(Z, \phi, \delta)\right],
\end{aligned} \tag{13}
$$

where $\nabla_\phi Z = 0$ and $\nabla_\theta Z = \nabla_\theta \mathcal{S}_\theta^{-1}(H)|_{H=\mathcal{S}_\theta(Z)}$ can be obtained by implicit differentiation of $S_\theta(Z) = H$ which results in $\nabla_\theta Z = -(\nabla_z \mathcal{S}_\theta(Z))^{-1}\nabla_\theta \mathcal{S}_\theta(Z)$. So for instance $\frac{\partial Z_1}{\partial \alpha_1} = -\frac{1}{p_{\alpha_1}(Z_1)}\frac{\partial \gamma(Z_1, \alpha_1)}{\partial \alpha_1}$, with $p_{\alpha_1}$ being the density function of $Z_1$ and recalling that $\theta = (a, b, \alpha_1, \ldots \alpha_d)$. We can thus optimize the variational bound using stochastic gradient descent with unbiased samples from (13). We remark that for instance in tensorflow probability [9], such implicit gradients are used by default as long as one simulates from the copula-like density using Proposition 1, implements the density function $c_\theta$ from (6) and applies the bijective transformations according to Algorithm 1. In this case, optimization using the proposed density proceeds analogously as if one would use any reparametrisable variational family such as Gaussian distributions.

## D   Additional details for Bayesian Neural Networks with Structured Priors

In the MNIST experiments, we train the network on 50000 training points out of 60000 and report the prediction error rates for the test set of 10000 images. We used a batch-size of 200 and used 4 Monte Carlo samples to compute the gradients during training and 100 Monte Carlo samples for the prediction on the test set. We used Adam with a learning rate in $\{0.0005, 0.0002\}$ for 20000 iterations. The hyper-parameter for the Horseshoe prior were $\nu = 4$, $s = 1$, so $c \sim \mathcal{IG}(2, 8)$, corresponding to a $t_4(0, 2^2)$ slab. Furthermore, for the global shrinkage factor, we have used $b_\tau \in \{0.1, 1\}$. The variational parameters of the copula-like density are restricted to be positive and we have defined them as the softmax: $x \mapsto \log(\exp(x) + 1)$ of unconstrained parameters, initialised so that softmax$^{-1}(\alpha_i) \sim \mathcal{N}(2, .01)$, softmax$^{-1}(a) = 15$ and softmax$^{-1}(b) = 2$. We have sampled $\delta$ according to (8) and initialised $\nu_i \sim \mathcal{U}(-0.2, 0.2)$ and the log-standard deviations of the marginal-like distribution as $\log \sigma_i = -3$. We aimed for an initial mean of 0 for $\beta_i^l$ and of $-3$ for the log of the remaining variables. We therefore choose $\mu_i$ so that the quantile of an initial Monte Carlo estimate for the mean of $V_i$ has the desired initial mean.

# E  Additional results for Bayesian Neural Networks with Gaussian Priors

Table 6: Variational approximations with transformations and different base distributions. Test log-likelihood for UCI regression datasets. Standard errors in parenthesis.

|         | Copula-like with rotation | Independent copula with rotation | Copula-like with IAF | Independent copula with IAF |
|---------|---------------------------|----------------------------------|----------------------|-----------------------------|
| Boston  | -2.85 (0.07)              | -2.84 (0.09)                     | -2.78 (0.1)          | -2.88 (0.09)                |
| Concrete| -3.29 (0.03)              | -3.30 (0.02)                     | -3.22 (0.02)         | -3.26 (0.02)                |
| Energy  | -1.04 (0.02)              | -2.34 (0.05)                     | **-0.93 (0.03)**     | -1.78 (0.07)                |
| Kin8nm  | 1.08 (0.01)               | 1.07 (0.01)                      | **1.10 (0.01)**      | 1.03 (0.01)                 |
| Naval   | 5.74 (0.05)               | 5.23 (0.05)                      | **5.97 (0.05)**      | 5.01 (0.05)                 |
| Power   | -2.82 (0.01)              | -2.85 (0.04)                     | -2.83 (0.04)         | -2.85 (0.01)                |
| Wine    | -1.01 (0.01)              | -1.02 (0.02)                     | -1.02 (0.02)         | -1.02 (0.02)                |
| Yacht   | -2.01 (0.04)              | -2.03 (0.06)                     | -1.69 (0.06)         | -1.94 (0.07)                |
| Protein | **-2.87 (0.00)**          | -2.94 (0.00)                     | -2.90 (0.01)         | -2.93 (0.01)                |

Table 7: Copula-like variational approximation without rotations and benchmark results. Test log-likelihood for UCI regression datasets. Standard errors in parenthesis.

|         | Copula-like without rotation | Bayes-by-Backprop results from [47] | SLANG results from [47] | Dropout results from [47] |
|---------|------------------------------|-------------------------------------|-------------------------|---------------------------|
| Boston  | -2.79 (0.08)                 | -2.66 (0.06)                        | -2.58 (0.05)            | **-2.46 (0.06)**          |
| Concrete| -3.25 (0.03)                 | -3.25 (0.02)                        | -3.13 (0.03)            | **-3.04 (0.02)**          |
| Energy  | -1.00 (0.03)                 | -1.45 (0.02)                        | -1.12 (0.01)            | -1.99 (0.02)              |
| Kin8nm  | 1.09 (0.01)                  | 1.07 (0.00)                         | 1.06 (0.00)             | 0.95 (0.01)               |
| Naval   | 5.45 (0.12)                  | 4.61 (0.01)                         | 4.76 (0.00)             | 3.80 (0.01)               |
| Power   | -2.83 (0.01)                 | -2.86 (0.01)                        | -2.84 (0.01)            | **-2.80 (0.01)**          |
| Wine    | -1.02 (0.01)                 | -0.97 (0.01)                        | -0.97 (0.01)            | **-0.93 (0.01)**          |
| Yacht   | -1.92 (0.06)                 | **-1.56 (0.03)**                    | -1.88 (0.01)            | **-1.55 (0.03)**          |
| Protein | -2.89 (0.01)                 | NA                                  | NA                      | **-2.87 (0.01)**          |