[Reviews · NeurIPS 2019]

Reviewer 1



After rebuttal ------------------------ Thank you for the rebuttal. After having read all the reviews and the rebuttal, I will keep my score of 7. That being said, I reviewer with Reviewer #3 that the experimental part can be improved and I still think the authors could do a better job analyzing and explaining the representative power and flexibility of the proposed variational family. Summary -------------------- The paper is proposing a new variational family for variational inference as an alternative to classic mean-field Gaussians and full-rank Gaussians. The new variational family is constructed as follows. The starting point is a copula-like density, meaning a density that lives on the hypercube, but do not have uniform marginals. The authors then propose to apply a normalizing flow-type map and a sequence of Given rotations to the random variable in order to make the resulting density more expressive. The authors present an algorithm to sample from the proposed distribution, which is meant as a plug in-replace for Gaussians. Finally, the paper is concluded with a set of numerical experiments Clarity ----------------- Overall, the paper is well written and well structured. Quality ------------------ The paper appears technically correct. The claims of the paper are supported to some degree by empirical evidence, but there is no theoretical analysis. Based on the simple 2D experiments 8 (Sec. 6.1 & 6.2), it is seen that the proposed method is capable of modelling asymmetric posterior distributions. The code is not included in the submission. Significance ---------------- The proposed construction is indeed interesting. This work would be of interest for both researchers and practitioners. Originality ----------- To the best of my knowledge, both the idea of using the distribution in eq. (6) as a base distribution and the idea of using Given rotations in variational inference are also novel. Other comments ------------- It is obvious that the density given in eq. (6) is a density? Figure 3: The quality of the figure is very bad on print

Reviewer 2



# Summary This paper presents a new family of variational distributions with posterior dependence preserved, targeting for high-dimensional problems. The construction of variational distribution is motivated by Sklar's theorem, in which the dependence structures and univariate margins are handled separately. While the marginal distribution of any copula has to be uniform, the proposed copula-like density allows for non-uniform marginal distributions. As a result, the dependency structure can be parametrized flexibly with linear complexity, and sampling from the copula-like density is easy. While the number of parameters in an unconstrained Gaussian covariance is quadratic. It has been pointed out by Opper, M., & Archambeau, C. (2009) that the number of parameters in Gaussian variational inference can be O(d) instead of O(d^2), with a parameterized covariance matrix. Please refer to Khan, M. E. et al. (2013) and the references therein. Admittedly, although the Gaussian covariance matrix is flexible and easier to optimize, it may not necessarily close to the true posterior as well. The dependency parameters presented in this paper are hard to optimize. To alleviate this problem, a sequence of Givens rotation as normalizing flow are considered to make further adjustments. It is nice to see that the transformation follows the FFT-style butterfly-architecture, and it could serve as standardization and potentially improve Variational Gaussian approximations as well. Experimental results demonstrate that the proposed method yields higher ELBO than mean-field Gaussian and Full-covariance Gaussian methods in several tasks. Potentially, the ELBO can be further improved by adopting a more flexible margin. # Originality This paper presents a novel way of constructing variational distributions motivated by Sklar's theorem. Toward high-dimensional dependence, both the copula-like density and the sparse rotation matrix are carefully designed, leading to a tradeoff between flexibility and computational efficiency. # Quality The mathematical derivations look solid. The experiments are well-designed and results are clearly presented. # Clarity This presentation is clear and the paper is well-organized. It would be good to add more discussion on the intuitions of the copula-like density before throwing out the complex mathematical form. More discussions on the efficiency of sampling and how the numerical value of delta manifests the strength of dependence are needed. # Significance Existing multivariate copulas are often restrictive. This paper breaks the constraint of uniform margins in copula density in exchange for better scalability of structured stochastic variational inference to high-dimensional problems. The method is appealing as a tool for approximate Bayesian inference in deep models.

Reviewer 3



The paper raises a copula-like variational distribution with rotation. The approach seems to work theoretically but the author should offer more detailed empirical evidence. Here are several major comments: 1. Since the proposed copula-like construction is a composition of multiple components, an ablation analysis is helpful in identifying the source of representative power. I am particularly interested in the performance of copula-like densities without rotation, copula-like densities with (any other) normalizing flow and mean-field Gaussian with rotation under the experimental setting in the paper. 2. What is the role of the transformation H defined in (7)? Since $\delta$ is fixed, H doesn’t improve the representative power of the variational family. 3. The comparison in Table 4 is somehow unfair. The author should provide the prediction errors for a 400*400 networks with a Gaussian prior using copula-like variational approximation. 4. In related work the author mentioned that their technique can be applied in high dimensions. But the experiments show that the rotation trick is not used in Bayesian neural network. The computational complexity of $O(d\log d)$ seems too high to deal with Bayesian neural networks. In addition, Figure 3 is barely readable and (7) seems to lack brackets.

[Author Response · NeurIPS 2019]

*We thank all the three reviewers for their constructive feedback. Please find our answers to major questions raised. Other points will be dealt with in the revised version. Code will be made available by the camera-ready deadline.*

Intuitions and flexibility of the proposed approach (Reviewer #1 and #2): We agree that we should have provided more intuition for the newly introduced density in the main document. The density builds up on Beta distributions as they are the marginals of the Dirichlet distributed random variable $U \sim \text{Dir}(\alpha)$. We then multiply $U$ with an independent random variable $G \sim \text{Beta}(a, b)$. The resulting random variable $Y = UG$ follows a Beta-Liouville distribution, which allows to account for negative dependence, inherited from the Dirichlet distribution through a Beta stick-breaking construction, as well as positive dependence via a common Beta-factor. Our contribution is now how to transform this distribution that lives within the simplex to one that has support on the full hypercube, while also allowing for efficient sampling and log-density evaluations. This discussion will be added to the main document to improve its clarity. We want to add that it is not obvious that eq. (6) is a density, but this follows from the proof of Proposition 1 by taking $f = 1$ in eq. (9) therein. Fig. 1 and 2 shows that our proposed family is quite flexible and expressive while being cheap to sample from. However, we will also try to include a more detailed numerical study of the proposed family of densities. Eventually, the flexibility of the variational approximation can be increased using different complementary work. We have illustrated in Section 6.2 that one can use a mixture of copula-like densities, that can also enhance the flexibility of the marginal distribution. Similarly, one could use the new density within a semi-implicit variational framework whose parameters are the output of a neural network conditional on some latent mixing variable.

Predefined budget of dependency parameters (Reviewer #2): Thank you for pointing out this very interesting question. We think that this point is relevant but would require a lot of ideas and results which are out the scope of the document and would distract the reader from the main ideas suggested in the paper.

Clarification of the transformation $\mathscr{H}$ (Reviewer #2 and #3): The main reason to consider $U = \mathscr{H}(V)$ with $V \sim c_\theta$ was 1) numerical stability since we need to compute quantile functions only on the interval $[\epsilon, 1 - \epsilon]$ using this transformation 2) to increase the flexibility of our proposed family. We suggest to take initially at random $\delta \in [0, 1]^d$ for the transformation $\mathscr{H}$ such that $\mathbb{P}(\delta_i = \epsilon) = p$ and $\mathbb{P}(\delta_i = 1 - \epsilon) = 1 - p$ with $p, \epsilon \in (0, 1)$ (in our experiments $\epsilon = 0.01$ and $p = 1/2$). We found that choosing a different (large enough) value of $\epsilon$ tends to yield no large difference, as this choice will get balanced by a different value of the standard deviation of the Gaussian marginal transformation. However, we observed that the parameter $p$ can impact the representative power of the variational distribution and the best and most sensible choice was $p = 1/2$ since it leads to a balanced proportion of components of $\delta$ equal to $\epsilon$ and $1 - \epsilon$. A more detailed discussion will be added to the document on this point.

Ablation studies and different normalizing flows (Reviewer #1 and #3): We remark that using a Gaussian mean-field distribution with rotations basically yields still a Gaussian approximation, assuming that the small effect of the transformation $\mathscr{H}$ can be neglected. For $X = \mathcal{O}X'$ with a mean-field distribution $X' \sim \mathcal{N}(\mu, \Lambda)$ and some rotation matrix $\mathcal{O}$, we have $X \sim \mathcal{N}(\mathcal{O}\mu, \mathcal{O}\Lambda\mathcal{O}^\top)$. We have shown in sections 6.1 and 6.2 that using a copula-like density instead of an independent-copula base distribution can be beneficial, as the latter corresponds to the results for the full-rank Gaussian case with one rotation in dimension $d = 2$. Following the reviews, we have also performed an analogous study for the BNNs with the basic message being that the transformation $\mathscr{H}$ is essential for the copula-like density, whereas application of additional rotations yields only a smaller improvement. Rotating a mean-field Gaussian also performed less competitively. We will include more details in the revised version and for brevity refer to the MNIST results in the next section. Replacing the rotations with non-linear transformations like Inverse Autoregressive Flows would be an interesting idea to explore further, but we did not have the chance to explore this before the deadline.

High-dimensional BNN and the rotation trick (Reviewer #1 and #3): The variational density for the BNNs includes the rotations. We have not stated this explicitly and this might have caused confusion as we also referred to the density without the rotations as copula-like in the ablation study for the logistic regression. We will make the appropriate correction in the revised document. For the considered BNN with 200k latent variables, the complexity increases by a factor of around 13 compared to a mean-field model. We expect that more ad-hoc tricks can be used to adjust the rotations to some computational budget. For instance, one could consider the series of sparse rotations $\mathcal{O}_1, \cdots, \mathcal{O}_k$, but with $2^k < d$, thereby allowing for rotations of the more adjacent latent variables only. We acknowledge that the format and size Figure 3 is not suitable. We plan to illustrate our result using only one figure presenting $\text{Cor}(\text{vec}(W^3_{\cdot,0}, W^3_{\cdot,1}))$ (corresponding to Fig. 3(a)) with better colors and with not all the neurons. We will also present our results by making bigger and more designed plots in the supplementary material. The motivation for the model in the last experiment was to illustrate that the proposed approach can be used in high-dimensional structured Bayesian models without having to specify more model-specific dependency assumptions in the variatonal family. Our experimental MNIST results show that for this given model, a copula-like approach can perform better (error rate 1.70% with rotations and 1.78% without) than a Gaussian family (3.4% with rotations but without $\mathscr{H}$; 3.82% in the mean-field; a low-rank covariance structure did not converge in our implementation). The proposed density seems a useful alternative to a Gaussian one for the considered model and we have not analysed if it compares favourably in other models mentioned in Table 4, which has been included to indicate that the prediction errors are roughly in line with current work for fully connected networks.

[Meta-Review · NeurIPS 2019]

The paper proposes a new variational family based on copula-like construction, with efficient sampling from the proposed copula-like density. The framework provides a new way of modeling multivariate dependency structure with a tradeoff between parsimony and flexibility. The reviewers are in general agreement that the technical contributions are adequate, but the authors are encouraged to analyze and explain the representative power and flexibility of the proposed variational family in greater detail and depth.